# Characteristics and Outcomes of Patients with *Delftia acidovorans* Infections: a Retrospective Cohort Study

Signe Marie Mehl Højgaard,[a] Omid Rezahosseini,[a] Jenny Dahl Knudsen,[b] Natascha Josephine Ulstrand Fuglebjerg,[c] Marianne Skov,[d] Susanne Dam Nielsen,[a,e] Zitta Barrella Harboe[a,c,e]

[a]Viro-immunology Research Unit, Department of Infectious Diseases, Rigshospitalet, University of Copenhagen, Copenhagen, Denmark

[b]Department of Clinical Microbiology, Rigshospitalet, University of Copenhagen, Copenhagen, Denmark

[c]Department of Pulmonary Medicine and Infectious Diseases, Copenhagen University Hospital, North Zealand, Denmark

[d]CF Centre Copenhagen, Paediatric Pulmonary Service, Department of Paediatrics and Adolescent Medicine and Department of Infectious Diseases, Copenhagen University Hospital, Rigshospitalet, Copenhagen, Denmark

[e]Department of Clinical Medicine, University of Copenhagen, Copenhagen, Denmark

Signe Marie Mehl Højgaard and Omid Rezahosseini contributed equally to this article. Authors order was determined according to the contribution.

**ABSTRACT** *Delftia acidovorans* (*D. acidovorans*) is a Gram-negative bacteria and an uncommon cause of human infections. This retrospective cohort study investigated clinical and microbiological characteristics and outcomes of patients with *D. acidovorans* infections. We included patients with culture-confirmed *D. acidovorans* infections attending Rigshospitalet, during 2002-2020. Fifty-nine patients with a median interquartile ranges (IQR) age of 47 (15-67) years were included. Thirty-five (59%) were males, and 57 (97%) had at least one comorbidity, including 25 (42%) with solid or hematologic malignancies. Eight (14%) were admitted to ICU, and 15 (25%) died within 365 days after infection. Persistent infection was found in 4 (6.8%) patients, and 41 (70%) had polymicrobial cultures, mainly with *Pseudomonas* spp. and *Stenotrophomonas maltophilia*. More than 85% of the *D. acidovorans* isolates were susceptible to meropenem or ceftazidime. Although, 88% and 62% of the isolates were resistant to gentamicin and colistin, respectively. *D. acidovorans* infections mainly affect patients with preexisting comorbidities, including malignancies. In the first year, all-cause mortality is considerable, polymicrobial cultures are common, and meropenem or cephalosporins with antipseudomonal activity could be the antibiotics of choice.

**IMPORTANCE** *Delftia acidovorans* (*D. acidovorans*) is a Gram-negative bacteria that can cause infection in immunocompetent and immunocompromised individuals. The current knowledge comes mainly from case reports and case series. In this retrospective cohort study, we found that *D. acidovorans* infections mainly affect male patients with preexisting comorbidities, including malignancies. Persistent infections were not common, and most of the patients had polymicrobial cultures, mainly with *Pseudomonas* spp. and *Stenotrophomonas maltophilia*. More than 85% of the *D. acidovorans* isolates were susceptible to meropenem or ceftazidime. In contrast, 88% and 62% of the isolates were resistant to gentamicin and colistin, respectively.

**KEYWORDS** *Comamonas acidovorans*, *Delftia acidovorans*, *Pseudomonas acidovorans*, intensive care units, microbial sensitivity test, mortality

Address correspondence to Zitta Barrella Harboe, zitta.barrella.harboe@regionh.dk, or Signe Marie Mehl Højgaard, signe.marie.mehl.hoejgaard@regionh.dk.

The authors declare a conflict of interest. O.R. received a grant from Rigshospitalet related to this work, and a grant from A. P. Møller Fonden not related to this work. S.D.N. received a grant from the Novo Nordisk Fonden. Z.B.H. received a grant from Independent Research Denmark. All other authors report no potential conflicts of interest.

**D**elftia acidovorans (*D. acidovorans*) is a Gram-negative, aerobe, rod shaped bacteria. It was previously named *Pseudomonas acidovorans* and *Comamonas acidovorans* (1). However, after sequencing of the bacteria's 16s rRNA in 1999, it was renamed to *D. acidovorans* (2). *D. acidovorans* is an environmental bacteria found in water and soil (3), which can also form biofilms (4). *D. acidovorans* is rarely a human pathogen; however, it has

**TABLE 1** Characteristics of the patients with *Delftia acidovorans* infection

| Characteristics | Vital status 365 days after the first positive *Delftia acidovorans* culture | | Total (*n* = 59) |
| --- | --- | --- | --- |
| | Alive (*n* = 44) | Deceased (*n* = 15) | |
| Age at the time of infection, median [IQR] | 45 [14; 69] | 61 [43; 69] | 47 [15; 67] |
| Gender, *n* (%) | | | |
| Male | 23 (52) | 12 (80) | 35 (59) |
| Cultured specimen, *n* (%) | | | |
| Blood | 14 (32) | 3 (20) | 17 (29) |
| Urine | 5 (11) | 3 (20) | 8 (14) |
| Airway secretions | 17 (39) | 3 (20) | 20 (34) |
| Tissue or wound | 4 (9) | 3 (20) | 7 (12) |
| Medical equipment or others | 4 (9) | 3 (20) | 7 (12) |
| Presence of concomitant microorganisms, *n* (%) | | | |
| Yes | 31 (70) | 10 (67) | 41 (70) |
| At least one comorbidity at the time of infection, *n* (%) | | | |
| Yes | 42 (96) | 15 (100) | 55 (93) |
| At least one re-admission within the first yr after infection, *n* (%) | | | |
| Yes | 28 (64) | 9 (60) | 37 (63) |
| ICU admissions within the first yr after infection, *n* (%) | | | |
| Yes | 5 (11) | 3 (6.8) | 8 (14) |

been described as the cause of infection in both immunocompetent (5, 6), and in immunocompromised individuals. *D. acidovorans* has been described as a causative agent in several clinical syndromes including pneumonia, urinary tract infections, empyema, endocarditis, peritonitis, catheter-related infections and bacteremia (7). Furthermore, it has been described as a cause of nosocomial infections (8, 9).

There is limited data about *D. acidovorans*, and current knowledge comes mainly from case reports and a retrospective case series including a few patients (7, 10). We investigated characteristics, antimicrobial susceptibility patterns and related hospitalization, ICU admission, and all-cause mortality after infection with *D. acidovorans* at a tertiary referral center in Copenhagen from 2002 to 2020.

## RESULTS

**Patient characteristics.** Out of 85 patients with at least one positive culture for *D. acidovorans*, 24 declined to participate, and two were excluded due to loss of follow-up. In total, 59 patients (69%) were included in the study.

Patient characteristics are presented in Table 1. In total, 35 patients (59%) were male, and the median interquartile ranges (IQR) age at the time of first episode of infection was 47 (15–67) years. The population consisted of 17 (29%) children, 26 (44%) adults, and 16 (27%) elderly patients.

**Comorbidities.** Two (3.4%) patients did not have any of the comorbidities at the time of *D. acidovorans* infection. The first patient was an adult male, and the specimen was collected from a wound. The second patient was an elderly female, and the sample was collected from a central venous catheter. No more information about comorbidities for the two patients was available. Thirty-seven (63%) had one and 20 (34%) had two or

more comorbidities. Twenty-five (42%) out of the 59 patients had cancer (10 solid cancers, 15 hematological malignancies), and 12 (20%) had cardiovascular disease. Five (8.5%) patients had congenital syndromes, and all were children (Table S1).

**Microbiological characteristics.** Forty-six (78%) out of the 59 patients had only one positive culture. Thirteen (22%) had two or more positive cultures, although none had samples from multiple sites. Considering the first positive culture, 17 (29%) specimens were blood, 20 (34%) were airway secretions, eight (14%) were urine, seven (12%) were wound or tissue, and seven (12%) were from other sites.

Four (6.8%) out of the 59 patients fulfilled the criteria for persisting infection or colonization, and two of the four were children. The first patient was a 65-year-old male with multiple comorbidities. The samples were taken from a wound due to ischemia on his foot. The patient died one month after the first positive culture. The second patient was a 66-year-old male with COPD, and samples were from airway secretions. The third patient was a 10-year-old child that was intubated due to neurologic disease and samples were tracheal secretions. The fourth case was a 1-year-old infant with liver disease, and the samples were from blood. The second, third, and forth patients were alive up to one year after infection.

*D. acidovorans* was found as monoculture in 18 (30%) out of the 59 specimens. Forty-one (70%) specimens were positive for more than one microorganism, mainly *Pseudomonas* sp. ($n = 12$), *Stenotrophomonas maltophilia* ($n = 10$), *Staphylococcus* sp. ($n = 8$), and yeast ($n = 8$).

More than 52, 51, 49, 49, and 47 of the cultured *D. acidovorans* were susceptible to meropenem, ceftazidime, imipenem, ciprofloxacin, and piperacillin-tazobactam, respectively; while 37, 36, and 32 were resistant to tobramycin, gentamicin, and colistin, respectively (Table 2).

Thirty-five (59%) out of the 59 patients had complete information about previous antibiotic therapy. Twenty-five (71%) out of the 35 received at least one antibiotic within 3 months before the first positive *D. acidovorans* culture. Penicillins and fluoroquinolones were patients' most common antibiotics before a positive *D. acidovorans* culture.

After a positive *D. acidovorans* culture, 23 (66%) out of the 35 patients received at least one new antibiotic. Fluoroquinolones and meropenem were the antibiotics most prescribed after a positive *D. acidovorans* culture (Table S2).

**Outcomes.** Thirty-one (53%) out of the 59 patients were admitted to the hospital at the time of positive culture. Six (10%) were admitted from more than 30 days before the positive culture. Twelve (20%) were admitted within 1 to 30 days before, and 14 (24%) patients were admitted on the day of positive culture. None of the patients were admitted to ICU at the same time of the positive culture.

Zero, 5 (8.5%), and 8 (14%) out of the 59 patients were admitted to ICU within 30, 180, and 365 days after the first positive culture, respectively.

Four out of the 59 patients (7%) died within 30 days after the first positive culture, 10 (17%) died within 180 days, and 15 (25%) patients died within the 365 days after the first positive culture. Only one patient died after having been admitted to ICU. Among the four patients who died within 30 days after the first positive culture, the specimens were collected from airway secretions ($n = 1$), tissue or wound ($n = 2$), and medical equipment ($n = 1$). Moreover, three out of the four had polymicrobial cultures.

## DISCUSSION

To our knowledge, this is the largest cohort study characterizing consecutive patients diagnosed with *D. acidovorans* infections. The patient population was heterogeneous, but most patients (97%) had at least one underlying comorbidity. All-cause mortality within the first year after infection was high, reaching one-fourth of the patients.

*D. acidovorans* infections were detected in all age groups, while 44% of patients were adults. Interestingly, 11 (48%) out of the 23 cases reported in the case series from 2015 were in the same age group (7).

Although *D. acidovorans* can affect immunocompetent individuals (6, 11, 12), most reported cases have had comorbidities or were immunocompromised (7). This is in line with our findings, where 97% of the patients had at least one comorbidity.

**TABLE 2** Antimicrobial susceptibility pattern of *Delftia acidovorans*[a]

| Susceptibility | Penicillin | Ampicillin | Mecillinam | AMC | TZP | Cefuroxime | Ceftriaxone | Cefpodoxime | Ceftazidime | Sulfamethizol | Trimethoprim | TMP-SMX |
|---|---|---|---|---|---|---|---|---|---|---|---|---|
| Resistant | 45 | 50 | 18 | <3 | <3 | 21 | 8 | 3 | <3 | 4 | 19 | <3 |
| Intermediate | * | <3 | * | <3 | 7 | 17 | 18 | * | >3 | * | 16 | * |
| Sensitive | * | <3 | <3 | 8 | 47 | 14 | 18 | 8 | 51 | 42 | 13 | 32 |
| Not Tested | 14 | 7 | >39 | 48 | <5 | 7 | 15 | 48 | 3 | 13 | 11 | >23 |

[a]The term Intermediate may represent a nonsusceptible phenotype. Not all isolates were tested against all antibiotics. Although, 36 (88%) out of 41 and 32 (62%) out of 52 isolates were resistant to gentamicin and colistin, respectively. AMC, Amoxicillin/clavulanic acid; TMP-SMX, Trimethoprim-Sulfamethoxazole; TZP, Piperacillin/Tazobactam; *, Not reported.

**TABLE 2** (Continued)

| Gentamicin | Tobramycin | Tetracycline | Ciprofloxacin | Moxifloxacin | Nitrofurantoin | Imipenem | Meropenem | Chloramphenicol | Aztreonam | Colistin |
|---|---|---|---|---|---|---|---|---|---|---|
| 36 | 37 | <3 | >5 | <3 | 4 | <3 | 3 | <3 | 9 | 32 |
| 4 | 14 | <3 | 3 | <3 | 3 | 3 | * | * | 29 | 13 |
| <3 | 4 | 29 | 49 | 8 | <3 | 49 | 53 | 3 | 16 | 7 |
| >17 | 4 | 27 | <3 | 48 | >48 | >4 | 3 | >49 | 5 | 7 |

*D. acidovorans* was mainly found in airway secretions (34%) and blood (29%), while 70% of the cultures were positive for another microorganism, most commonly *Pseudomonas* sp. and *Stenotrophomonas maltophilia*. More than 70% of patients received at least one antibiotic within 3 months before the first positive *D. acidovorans and* Penicillins and fluoroquinolones were the most common ones.

Information about antimicrobial susceptibility of uncommon Gram-negative and nonfermenting bacteria could be of clinical importance (13). In our cohort, *D. acidovorans* showed antibiotic resistance to some antibiotics commonly used to treat Gram-negative infections. We found that meropenem or ceftazidime could be suitable choices for empirical treatment, while isolates mainly were resistant to gentamicin and colistin, whereas both antibiotics are commonly used for their antipseudomonal activity (14). It is worth mentioning that antimicrobial susceptibility patterns could be different in other centers. Moreover, concomitant microorganisms should be considered before the selection of antibiotics.

*D. acidovorans* colonization or persisting infection was found in a relatively small proportion, in about 7% of patients. Although, this is lower than proportions of persistent infection with other species in the order Burkholderiales, such as *Achromobacter* and *Alcaligenes* that reach to more than 30% (15, 16). Unfortunately, there was no previous publication about *D. acidovorans* for comparison.

We found that 14% of the patients were admitted to the ICU, and 25% died within 365 days after the infection. This could be related to the comorbidities or coinfections that our patients had, and we cannot discuss causality.

Our study had some strengths and limitations. We aimed to include all consecutive patients with *D. acidovorans* infection, regardless of age and preexisting condition. Moreover, we had complete information on death and laboratory information on antimicrobial susceptibility. We obtained permission to review clinical charts from deceased patients, but alive patients should provide written consent, and not all of them accepted to participate. Therefore, we had some degree of bias in selecting cases, with a potential to overestimate the contribution of comorbidities in the population described. We also had to address missing data, as not all historic patient's charts and deceased patient's charts were available to access digitally via electronic records. Furthermore, using current data, it was not possible to differentiate between colonization and infection. Therefore, it is possible that some patients had colonizations without clinical importance. Changes in susceptibility testing methods and, possibly, breakpoints could influence the susceptibility results. Although everything was done according to the standard of care and methods were calibrated if necessary.

In conclusion, we characterized a large cohort of consecutive patients who presented with *D. acidovorans* infections. *D. acidovorans* infections mainly affect male patients with preexisting comorbidities, including malignancies. The first-year all-cause mortality is considerable, and polymicrobial cultures are common, and meropenem or cephalosporins with antipseudomonal activity could be the antibiotics of choice.

## MATERIALS AND METHODS

**Settings and the scientific data approvals.** We conducted a retrospective cohort study from 8 January 2002 to 17 February 2020, at the Copenhagen University Hospital, Rigshospitalet, a tertiary referral hospital in the Capital Region. Rigshospitalet is the largest and a highly specialized hospital in Denmark with 1,182 beds that hosts most of the specialty departments with 12,000 personnel. In 2020, Rigshospitalet was responsible for 77,946 inpatient visits and 1,130,040 outpatient visits (17).

Patients were identified from laboratory records at the Department of Clinical Microbiology. Approval for the study was obtained from the Centre for Regional Development (jr.nr. R-20042314); the Copenhagen University Hospital, Rigshospitalet (jr.nr WZ: 21014838), and the Knowledge Center for Data Reviews (jr.nr. P-2021-271). According to Danish law, approval from an ethical committee was not required.

**Patient population.** Positive cultures with *D. acidovorans* (or any of the previous taxonomic names including *Comamonas-* or *Pseudomonas acidovorans*) were identified via the local laboratory information system hosted by the Department of Clinical Microbiology. *D. acidovorans* infection was defined as the growth of *D. acidovorans* in clinical samples obtained from a patient either admitted to the hospital or attending an outpatient consultation. Patients with multiple positive samples were included only once.

Patients with more than one positive culture from the same site, we defined "persisting infection" or "colonization."

In Denmark, each person has a unique Central Person Register (CPR) number, and vital status is registered in the Danish Civil Registration System using their CPR number (18). All patients who were alive at the initiation of the study were contacted with an invitation letter containing information about the project, a contact email address for questions, and a letter of consent. The patients were asked to sign this and return the consent to the Department of Infectious Diseases if they would like to participate in the study. Nonresponders were contacted twice. Twenty-four patients who did not return a consent form after two contacts were registered as declining to participate in the study. According to the ethical approvals, all deceased patients were included in the study.

**Data sources and availability.** Patients' characteristics, information about comorbidities, hospitalization, ICU admission, death, and microbiological data were collected from electronic medical records and stored in a Research Electronic Data Capture (REDcap) database hosted at the Capital Region of Denmark (19). Starting from 2010, all information was retrieved from electronic medical records. Half of the patients ($n = 30$) had an infection before 2010, and the information was in some cases incomplete. The data are available on request from the corresponding author and are not publicly available due to Danish legislations.

**Definitions.** We defined children, adults, and the elderly as patients younger than 18, 18-65, and older than 65 years old, respectively.

We reported hospital admission, ICU admission, or death up to 365-days after the first sample date. Elective admissions, planned surgery, visits to outpatient clinics, or the emergency department without admission were not considered hospital admissions.

*D. acidovorans* infection was defined as the growth of *D. acidovorans* in clinical samples obtained from any site in a patient either admitted to the hospital or attending an outpatient consultation. Patients with multiple positive samples were included only once. For patients with more than one positive culture from the same site, we defined persisting infection or colonization. Persisting infection or colonization was defined as the diagnosis of *D. acidovorans* in a new culture obtained from a sample taken more than 14 days from the previous episode of infection, based on the definition of Repeated Infection Timeframe from the Centers for Disease Control and Prevention (CDC) (20).

**Comorbidities.** All possible congenital syndromes regardless of etiology, were considered as one diagnosis except for primary ciliary dyskinesia and cystic fibrosis, which were included as two separate diagnoses. Myocardial infarction, congestive heart failure, and peripheral vascular disease were considered as cardiovascular diseases. Other comorbidities include cerebral vascular disease, neuromuscular disease, connective tissue diseases, diabetes, idiopathic pulmonary hypertension, chronic obstructive pulmonary disease, interstitial lung disease, primary ciliary dyskinesia, cystic fibrosis, liver diseases, renal dysfunction, hematologic malignancies, solid tumors, bone marrow transplant recipients, and solid organ transplant recipient.

**Data on antimicrobial diagnostics and susceptibility testing.** Before 2010, the laboratory diagnosis of Gram-negative bacteria was based on conventional laboratory tests as Gram stain and fluid microscopy. A Gram-negative, motile rod, which was oxidase-positive and nitrate reactive; produced acid from mannitol but not from glucose, sucrose, lactose or maltose; and was negative in urease, arginine dihydrolase, ornithine, and lysine decarboxylases, was considered as *D. acidovorans*. After 2010, the laboratory diagnosis was additionally confirmed by using matrix-assisted laser desorption ionization-time to flight mass spectrometry (MALDI-TOF).

All susceptibility testing, identification, and other testing were done in accordance with routine laboratory practices. All the susceptibility testing were until 2016 done by Rosco tables (Rosco, Taastrup, Denmark) and/or E-tests (bioMérieux, France) on Danish Blood Agar plates (Statens Serum Institut, Denmark), using clinical and laboratory standard institute (CLSI) breakpoints. Later, EUCAST breakpoints (www EUCAST.org) using *Pseudomonas aeruginosa* or nonspecies related breakpoints using Oxoid Disks (ThermoFisher, UK) on Danish Blood agar plates were used.

**Statistical analysis.** Proportions are presented as percentages, and continuous data are presented as medians with IQR. Comparisons were calculated using the Mann-Whitney U test for continuous variables, and Fischer's Exacts test for categorical variables. Proportions for ICU admission and mortality were reported within 30, 180, and 365 days after the first episode of *D. acidovorans* infection. All analyses were conducted using R statistical software version 3.6.1 (21). *P* values $< 0.05$ were considered statistically significant.

## SUPPLEMENTAL MATERIAL

Supplemental material is available online only.
**SUPPLEMENTAL FILE 1**, PDF file, 0.1 MB.

## ACKNOWLEDGMENTS

We thank all patients who participated in this study.

S.M.M.H., O.R., J.D.K., N.J.U.F., S.D.N., and Z.B.H. designed the study. S.M.M.H., J.D.K., and M.S. collected the data. S.M.M.H., O.R., and Z.B.H. did statistical analyses. S.M.M.H., O.R., and Z.B.H. wrote the manuscript. All authors read, revised, and commented on the manuscript and approved the final version of the manuscript.

O.R. received a grant from Rigshospitalet related to this work, and a grant from A. P. Møller Fonden not related to this work. S.D.N. received a grant from the Novo Nordisk Fonden. Z.B.H. received a grant from Independent Research Denmark. All other authors report no potential conflicts of interest.

No external funding was received for this study.

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
