## [Reviewer comments · Microbiology Spectrum]

Microbiology Spectrum

Characteristics and Outcomes of Patients with *Delftia acidovorans* Infections: A Retrospective Cohort Study

Signe Højgaard, Omid Rezahosseini, Jenny Knudsen, Natacha Fuglebjerg, Marianne Skov, Susanne Nielsen, and Zitta Harboe

Corresponding Author(s): Zitta Harboe, Copenhagen University Hospital, North Zealand

Review Timeline:

Submission Date:	January 26, 2022
Editorial Decision:	April 5, 2022
Revision Received:	June 16, 2022
Accepted:	June 27, 2022

Editor: Krisztina Papp-Wallace

Reviewer(s): Disclosure of reviewer identity is with reference to reviewer comments included in decision letter(s). The following individuals involved in review of your submission have agreed to reveal their identity: Mohamed H. Yassin (Reviewer #2); Matthew P Crotty (Reviewer #5)

Transaction Report:

DOI: <https://doi.org/10.1128/spectrum.00326-22>

April 5, 2022

Dr. Zitta Barrella Harboe
Copenhagen University Hospital, North Zealand
Copenhagen
Denmark

Re: Spectrum00326-22 (Characteristics and Outcomes of Patients with Deltia acidovorans Infections: A Retrospective Cohort Study)

Dear Dr. Zitta Barrella Harboe:

Link Not Available

Sincerely,

Krisztina Papp-Wallace

Journals Department
Reviewer comments:

Reviewer #2 (Comments for the Author):

Thank you for writing this important manuscript. Overall the manuscript is well-written and very informative.

I have the following comments;

- 1) Abstract: no issues very good presentation of the work
- 2) Background: is concise and related to the manuscript aim
- 3) The data collected and the explanation of the Microbiology methods are very satisfying to feel comfortable with the data.
- 4) The results & graphs are clear and very informative

Reviewer #3 (Comments for the Author):

This is a paper describing the characteristics of patients with *D. acidovorans* infection at a single Danish hospital. The paper is generally well written. There are, however, a number of limitations. Notably, the paper does not adequately describe the nature of comorbid medical conditions and the treatment of patients included. These details are necessary for readers using this paper for any clinical application. Additionally, a multivariable analysis is not needed, particularly in light of other missing information. There are several other comments which should be addressed which are included below.

Methods

- Some additional information on the hospital would be of interest to help place the patient characteristics in context (e.g., bed number, services offered, referral area, etc.)
- Patient population: If relevant, were patients with positive cultures from multiple sites included only once? If any patients had *Delftia* isolated multiple times on different occasions, were these patients included once?
- Patient population: Did any living patients decline participation in the study? How was vital status ascertained? Did any potentially eligible patients not respond to the invitation? This is mentioned in the results but would specify here instead.
- Comorbidities: please clarify if all possible congenital syndromes, regardless of etiology, were considered as one diagnosis. Were any other comorbidities assessed or only those described? Primary ciliary dyskinesia is specifically mentioned, is this inclusive of cystic fibrosis?
- Microbiology: there are a potentially large number of organism meeting the pre-2010 criteria as only three biochemical reactions are described. Were more extensive biochemical characterizations beyond oxidase, nitrate, and mannitol performed in the clinical microbiology laboratory at this time? This does not seem sufficient to adequately identify an organism to the species level.
- Microbiology: was all susceptibility testing, identification, and other testing done in accordance with routine laboratory practices?
- Statistics: the use of a Cox regression model seems somewhat unnecessary in a descriptive epidemiology study of a single organism in a limited number of patients with no defined control group. Recommend removing inferential / multivariable models and providing simple descriptive statistics. This will not limit the potential relevance or impact of the study.

Results

- Comorbidities: Why were the 6 patients without comorbidities in the hospital? What were the culture sites and why were these obtained?
- Microbiologic characteristics: were the two children included in the four patients who fulfilled criteria for persisting infection / colonization?
- Microbiologic characteristics: did any patients have more than one positive culture?
- Microbiologic characteristics: please specify what antimicrobials were tested in what time period. Table 3 is not included in the paper for review.
- Outcomes: were any of the patients in the ICU at the time of culture? What was the timing of cultures relative to hospital admission?
- Survival analysis: again, this should be removed. The study is not adequately designed or powered for any inferential statistics.

General

- The descriptive characteristics of the patients are incredibly limited and do not help with understanding the patient population included. Standard items, such as relevant medical comorbidities (COPD, diabetes, etc.) in addition to more detailed information (solid organ transplantation, malignancy status, etc.) should be provided in Table 1. Additionally, microbiologic source and epidemiologic classification of the infection must be provided.
- The description of susceptibility results is very much appreciated. However, Figure 2 should provide information on the number of isolates tested for each drug and include all drugs tested - this appears to be a subset
- No information at all is provided on how the patients were treated. The sample size is small, but as mentioned, this is the largest study to date. Some information on treatment should be included.

Reviewer #4 (Comments for the Author):

The manuscript by Hojgaard et al provides an interesting look at the prevalence and clinical outcomes associated with *Delftia acidovorans* infections during a specific timeframe in a high resource setting. Although there are clear limitations to the retrospective study, the authors clearly outline these in their discussion. Some specific comments:
Lines 161-166: The manuscript comments on using both CLSI and EUCAST breakpoints for the organism; it is unclear as to why this was the case.

Line 196: Were all cases felt to represent infection, especially cultures isolated from superficial wounds? This reviewer has seen this organism in polymicrobial wounds and sputum cultures where it likely has represented a colonizer and not necessarily clinically significant.

With the small numbers included in the study, it would have been possible to perform a chart review to assess the cause of mortality in the cases that were deceased. How many cases can be reasonably attributed to *Delftia* infection?

Reviewer #5 (Comments for the Author):

General Comments

1. Evaluating the number of comorbidities is certainly of interest, however, "equal footing" for each of the comorbidities evaluated may not be optimal. Consideration to utilization of a more comprehensive score such as the Charlson or Elixhauser Comorbidity Indexes should be considered.
2. Due to the high number of polymicrobial infections (70%) differentiating the role/impact of *D. acidovorans* is extremely challenging. This will be a major limitation regardless of how the data is analyzed and presented. However, it would be helpful to the readership to recognize this as a limitation and attempt to be as detailed as possible when depicting the types of infections that were polymicrobial. For instance, more detailed summary of suspected source of polymicrobial bloodstream infections may assist in identifying opportunities for recognition and optimal treatment in similar patients. Moreover, the specific organisms identified concomitantly suggest a patient population that has likely had significant healthcare and possibly antibiotic exposure. Information on the hospitalization (previous or time from admission to culture collection) and any antibiotic exposure would add substantial value to understanding patients at risk for infections with *D. acidovorans*, please consider including if possible.
3. The finding of lower mortality among female patients may be of interest, however, context is needed. Is there biologic plausibility for such a finding? Related to infection type differences between females/males? Or is this simply a chance finding. Context in the discussion is needed if this is to remain an emphasized result.
4. Because mortality at 365 days could be attributable to numerous confounders, consideration of evaluating other clinical outcomes may be warranted. Evaluation of "clinical failure" or "infection-related mortality" may be of interest in more specifically evaluating the impact of the *D. acidovorans* infection on the mortality.

Specific Comments

1. P6 L125: Please consider specifying here whether after second contact non-responders were excluded (as it seems they were from results later in the manuscript).
2. P7 L150-154: Were transplant (solid organ) or other immunocompromising comorbidities or therapies considered?
3. P8 L161-166 and 19 Figure 2: Changes in testing methodology and possibly breakpoints utilized over time could influence the susceptibility results depicted. Please consider expressly identify the frequency in which testing modalities were utilized. Additionally, consideration for more directly sharing MIC results or what specific breakpoint was utilized for interpretation is needed to allow the readership to interpret such results in a clinical context.
4. P16-17 Tables 1 and 2 (and results section): Infection type (or specimen source) findings are interesting with lower mortality rate in bloodstream vs urinary source infections. Perhaps this relates to general comment #4 above (infection related mortality or due to comorbidity/confounder?). More detail surrounding the infection types and how they interact with the outcome (mortality) would seemingly be of utility to the readership.
5. P11 L237-238: This statement seems to lack context: is it compared to no infection or other pathogen, please consider providing more context.
6. P16 Table 1: Specifics on the individual comorbidities (e.g., COPD, CF) and outcomes would be of interest to readership, please consider including in the table.

Staff Comments:

Preparing Revision Guidelines

Please return the manuscript within 60 days; if you cannot complete the modification within this time period, please contact me. If you do not wish to modify the manuscript and prefer to submit it to another journal, please notify me of your decision

immediately so that the manuscript may be formally withdrawn from consideration by Microbiology Spectrum.

Dear Editor,

Thank you for allowing us to revise and improve our manuscript. We also thank the reviewers for carefully reading the manuscript and constructive remarks. Below is a point-by-point reply to the reviewers' comments. All references to the line and page are to the manuscript version with "track changes". In order to answer the issues raised, we have divided the comments into individual points. Please see the responses below.

Reviewer # had no comments. Thanks for this.

Reviewer #2 (Comments for the Author):

Thank you for writing this important manuscript. Overall the manuscript is well-written and very informative.

I have the following comments;

- 1) Abstract: no issues very good presentation of the work
- 2) Background: is concise and related to the manuscript aim
- 3) The data collected and the explanation of the Microbiology methods are very satisfying to feel comfortable with the data.
- 4) The results & graphs are clear and very informative

Response: Thank you for reviewing our manuscript.

Reviewer #3 (Comments for the Author):

This is a paper describing the characteristics of patients with D. acidovorans infection at a single Danish hospital. The paper is generally well written. There are, however, a number of limitations.

Reviewer #3, Comment 1

Notably, the paper does not adequately describe the nature of comorbid medical conditions and the treatment of patients included. These details are necessary for readers using this paper for any clinical application.

Response to Reviewer #3, Comment 1: Thank you for the comments. We agree that it would be preferable to have more information on comorbidities and treatment. Unfortunately, not all the data was available for collection. We have collected all available information and responded to your comments. Please see the following responses:

Reviewer #3, Comment 2

Additionally, a multivariable analysis is not needed, particularly in light of other missing information. There are several other comments which should be addressed which are included below.

Response to Reviewer #3, Comment 2: We agree with the argument and removed the multivariable analysis.

Reviewer #3, Comment 3

Methods

- Some additional information on the hospital would be of interest to help place the patient characteristics in context (e.g., bed number, services offered, referral area, etc.)

Response to Reviewer #3, Comment 3: Thank you for the important suggestion. We included more information about Rigshospitalet on page 4, lines 18-22, which read:

"We conducted a retrospective cohort study from January 8th, 2002, to February 17th, 2020, at the Copenhagen University Hospital, Rigshospitalet, a tertiary referral hospital in the Capital Region."

and now reads: *"We conducted a retrospective cohort study from January 8th, 2002, to February 17th, 2020, at the Copenhagen University Hospital, Rigshospitalet, a tertiary referral hospital in the Capital Region. Rigshospitalet is the largest and a highly specialized hospital in Denmark with 1182 beds and hosts most of the specialty departments with 12000 personnel. In 2020, Rigshospitalet was responsible for 77,946 inpatients and 1,130,040 outpatient visits (Ref)."*

Ref: Key figures. Selected key figures for 2020. <https://www.rigshospitalet.dk/english/about-us/Pages/key-figures.aspx>

Reviewer #3, Comment 4

- Patient population: If relevant, were patients with positive cultures from multiple sites included only once? If any patients had Delftia isolated multiple times on different occasions, were these patients included once?

Response to Reviewer #3, Comment 4: Yes, we did not have any patients with positive cultures from multiple sites, and patients with multiple positive samples were included only once. For patients with more than one positive culture from the same site, we defined "persisting infection or colonization". We added on page 6, lines 20-23, which read:

"D. acidovorans infection was defined as the growth of D. acidovorans in clinical samples obtained from a patient either admitted to the hospital or attending an outpatient consultation."

now it reads: *"D. acidovorans infection was defined as the growth of D. acidovorans in clinical samples obtained from any site in a patient either admitted to the hospital or attending an outpatient consultation. Patients with multiple positive samples were included only once. For patients with more than one positive culture from the same site, we defined "persisting infection or colonization."*

Furthermore, we added on page 7, lines 1-4, which read:

"We defined persisting infection or colonization as the diagnosis of D. acidovorans in a new culture obtained from a sample taken more than 14 days from the previous episode of infection, based on the definition of Repeated Infection Timeframe from the Centers for Disease Control and Prevention (CDC) [12]."

and now reads: "*Persisting infection or colonization was defined as the diagnosis of D. acidovorans in a new culture obtained from a sample taken more than 14 days from the previous episode of infection, based on the definition of Repeated Infection Timeframe from the Centers for Disease Control and Prevention (CDC) [12]*"

We also added on page 9, line 12-23, which read:

"Four (6.8%) out of the 59 patients fulfilled the persisting infection or colonization criteria, and two were children. Considering the first positive culture, 17 (29%) specimens were blood, 20 (34%) sputum, eight (14%) urine, seven (12%) wound or tissue, four (6.8%) from medical devices, and three (3.4%) specimens were not classified."

and now reads: "*Forty-six (78%) out of the 59 patients had only one positive culture. Thirteen (22%) had two or more positive cultures, although none had positive samples from multiple sites. Considering the first positive culture, 17 (29%) specimens were blood, 20 (34%) airway secretions, eight (14%) urine, seven (12%) wounds or tissue, and seven (12%) from other sites.*

Four (6.8%) out of the 59 patients fulfilled the criteria for persisting infection or colonization, and two of the four were children. The first patient was a 65-year-old male with multiple comorbidities. The samples were taken from the wound due to ischemia on his foot. The patient died one month after the first positive culture. The second patient was a 66-year-old male with COPD; samples were from airway secretions. The third patient was a 10-year-old child that was intubated due to neurologic disease, and samples were tracheal secretions. The fourth case was a 1-year-old infant with liver disease; the samples were from blood. The second, third, and fourth patients were alive up to one year after infection."

Reviewer #3, Comment 5:

- Patient population: Did any living patients decline participation in the study? How was vital status ascertained? Did any potentially eligible patients not respond to the invitation? This is mentioned in the results but would specify here instead.

Response to Reviewer #3, Comment 5: Thank you for this comment. Due to the ethical approvals, if patients did not return a consent form after two contacts, they were registered as declining to participate in the study. No living patients actively declined participation; however, 24 individuals did not return the consent. Vital status was ascertained via the Danish CPR registry. A Danish citizen's death will always be registered in the CPR registry.

To clarify, we added on page 5, lines 12-22, which read:

"All patients who were alive at the initiation of the study were contacted with an invitation letter containing information about the project, a contact email address for questions, and a letter of consent. The patients were asked to sign this and return the consent to the Department of Infectious Diseases if they would like to participate in the study. Non-responders were contacted

twice. According to the ethical approvals, all deceased patients' records were included in the study."

now it reads:

"In Denmark, each person has a unique Central Person Register (CPR) number, and vital status is registered in the Danish Civil Registration System (Ref 2). All patients who were alive at the initiation of the study were contacted with an invitation letter containing information about the project, a contact email address for questions, and a letter of consent. The patients were asked to sign this and return the consent to the Department of Infectious Diseases if they would like to participate in the study. Non-responders were contacted twice. Twenty-four patients who did not return a consent form after two contacts were registered as declining to participate in the study. According to the ethical approvals, all deceased patients were included in the study."

Ref2: Pedersen CB. The Danish Civil Registration System. *Scand J Public Health.* 2011 Jul;39(7 Suppl):22-5. doi: 10.1177/1403494810387965. PMID: 21775345.

Reviewer #3, Comment 6:

- Comorbidities: please clarify if all possible congenital syndromes, regardless of etiology, were considered as one diagnosis. Were any other comorbidities assessed or only those described? Primary ciliary dyskinesia is specifically mentioned, is this inclusive of cystic fibrosis?

Response to Reviewer #3, Comment 6: Thank you for raising these critical points. All congenital syndromes were included as a single category, except for primary ciliary dyskinesia and cystic fibrosis, which were two separate diagnoses. Rigshospitalet has a highly specialized function for children and adults with cystic fibrosis and primary ciliary dyskinesia, which may be why many patients with these diagnoses were included. We had six patients categorized as "without comorbidity," while they had comorbidities that were not mentioned in our list. Therefore, we added some new comorbidities to the list and revised them in the text. To clarify, we added on page 7, lines 5-12, which read:

"We categorized congenital syndromes as one comorbidity. Myocardial infarction, congestive heart failure, and peripheral vascular disease were considered cardiovascular diseases. Other comorbidities include cerebral vascular disease, connective tissue diseases, diabetes, chronic obstructive pulmonary disease, primary ciliary dyskinesia, cystic fibrosis, renal disease, hematologic malignancies, and solid tumors."

and now reads: *"All possible congenital syndromes regardless of etiology were considered as one diagnosis except for primary ciliary dyskinesia and cystic fibrosis which were included as two separate diagnoses. Myocardial infarction, congestive heart failure, and peripheral vascular disease were considered as cardiovascular diseases. Other comorbidities include cerebral vascular disease, neuromuscular disease, connective tissue diseases, diabetes, idiopathic pulmonary hypertension, chronic obstructive pulmonary disease, interstitial lung disease, primary ciliary dyskinesia, cystic fibrosis, liver diseases, renal dysfunction, hematologic malignancies, solid tumors, bone marrow transplant recipients, and solid organ transplant recipient."*

We also revised on page 10, lines 1-7, which read:

"Six patients (10%) did not have comorbidities at the time of D. acidovorans infection, while 33 (56%) had one and 20 (34%) had two or more comorbidities. Twenty-five (42 %) out of the 59 patients had cancer (10 solid cancers, 15 hematological malignancies), and 12 (20%) had cardiovascular disease (Table 2)."

now reads:

"Two (3.4%) patients did not have any of the comorbidities at the time of D. acidovorans infection. The first patient was an adult male, and the specimen was collected from a wound. The second patient was an elderly female, and the sample was collected from a central venous catheter. No more information about comorbidities for the two patients was available. Thirty-seven (63%) had one, and 20 (34%) had two or more comorbidities. Twenty-five (42 %) out of the 59 patients had cancer (10 solid cancers, 15 hematological malignancies), and 12 (20%) had cardiovascular disease. Five (8.5%) patients had congenital syndromes, and all were children. (Supplementary Table 1)."

Supplementary Table 1: Comorbidities identified at the time of *D. acidovorans* infection.

Comorbidities (alphabetically)	number
Bone marrow transplant recipient	<3
Cardiovascular disease	
Myocardial infarction	3
Congestive heart failure	4
Peripheral vascular disease	5
Cerebral vascular disease	7
Chronic obstructive pulmonary disease	4
Congenital syndromes	5
Connective tissue disease	3
Cystic fibrosis	4
Diabetes	3
Hematologic malignancies	15
Interstitial lung disease	<3
Liver disease	<3
Neuromuscular disease	<3
No known comorbidity	<3
Primary ciliary dyskinesia	6
Renal dysfunction	4
Solid cancer	10
Solid organ transplant recipient	<3

Reviewer #3, Comment 7:

- Microbiology: there are a potentially large number of organism meeting the pre-2010 criteria as only three biochemical reactions are described. Were more extensive biochemical characterizations beyond oxidase, nitrate, and mannitol performed in the clinical microbiology laboratory at this time? This does not seem sufficient to adequately identify an organism to the species level.

Response to Reviewer #3, Comment 7: Thank you for pointing out this important issue. Yes, before 2010, we used conventional diagnostic as routine identifications - a long line of tubes, read for up to 28 days. We had tests that should be negative to ensure the diagnosis. Therefore, we elaborated on page 7, lines 14-23, and page 8, lines 1-3, which read:

"Before 2010, the laboratory diagnosis of Gram-negative bacteria was based on conventional laboratory tests as Gram stain and fluid microscopy. A Gram-negative, motile rod, which was oxidase-positive, nitrate reactive, and produced acid from mannitol, was considered as D. acidovorans. After 2010, the laboratory diagnosis was additionally confirmed by using matrix-assisted laser desorption ionization-time to flight mass spectrometry (MALDI-TOF). Susceptibility testing was done by Rosco tables (Rosco, Taastrup, Denmark) and/or E-tests (bioMerieux, France) on Danish Blood Agar plates (Statens Serum Institut, Denmark), using classification by clinical and laboratory standard institute (CLSI) breakpoints. Moreover, EUCAST breakpoints (www EUCAST.org), using Pseudomonas aeruginosa or non-species related breakpoints using Oxoid Disks (ThermoFisher, UK) on Danish Blood agar plates were used."

and now reads: *"Before 2010, the laboratory diagnosis of Gram-negative bacteria was based on conventional laboratory tests as Gram stain and fluid microscopy. A Gram-negative, motile rod, which was oxidase-positive, nitrate reactive, and produced acid from mannitol but not from glucose, sucrose, lactose or maltose, was negative in urease, arginine dihydrolase, ornithine and lysine decarboxylases, was considered as D. acidovorans. After 2010, the laboratory diagnosis was additionally confirmed by using matrix-assisted laser desorption ionization-time to flight mass spectrometry (MALDI-TOF). All susceptibility testing, identification, and other testing were done in accordance with routine laboratory practices. All the susceptibility testing were until 2016 done by Rosco tables (Rosco, Taastrup, Denmark) and/or E-tests (bioMerieux, France) on Danish Blood Agar plates (Statens Serum Institut, Denmark), using clinical and laboratory standard institute (CLSI) breakpoints. Later, EUCAST breakpoints (www EUCAST.org), using Pseudomonas aeruginosa or non-species related breakpoints using Oxoid Disks (ThermoFisher, UK) on Danish Blood agar plates were used."*

Reviewer #3, Comment 8:

Microbiology: was all susceptibility testing, identification, and other testing done in accordance with routine laboratory practices?

Response to Reviewer #3, Comment 8: Yes, as you correctly pointed out, all susceptibility testing, identification, and other testing were made following routine laboratory practices. We added this on page 7, lines 21-22. Please see Response to Reviewer #3, Comment 7.

Reviewer #3, Comment 9:

- Statistics: the use of a Cox regression model seems somewhat unnecessary in a descriptive epidemiology study of a single organism in a limited number of patients with no defined control group. Recommend removing inferential / multivariable models and providing simple descriptive statistics. This will not limit the potential relevance or impact of the study.

Response to Reviewer #3, Comment 9: We agree with the argument and removed multivariable models and survival analyses. We revised on page 8, lines 8-10, which read:

"Proportions are presented as percentages, and continuous data are presented as medians with interquartile ranges (IQR). Comparisons were calculated using the Mann-Whitney U test for continuous variables, and Fischer's Exact test for categorical variables. Survival analysis and event probability plots were used to investigate the probability of survival. A Cox regression model was used to determine risk factors for death in patients infected with D. acidovorans. Baseline characteristics of patients (age, gender, comorbidities) were considered clinically relevant and possible risk factors for death. Patients were followed from a positive culture to death for survival analysis, 365-days after the first sample or December 31st, 2020, whichever came first. Hazard ratios (HR) were adjusted for gender and age group. All analyses were conducted using R statistical software version 3.6.1 [13]. P values < 0.05 were considered statistically significant."

now reads:

"Proportions are presented as percentages, and continuous data are presented as medians with interquartile ranges (IQR). Comparisons were calculated using the Mann-Whitney U test for continuous variables and Fischer's Exact test for categorical variables. Proportions for ICU admission and mortality were reported within 30, 180, and 365 days after the first episode of D. acidovorans infection. All analyses were conducted using R statistical software version 3.6.1 [13]. P values < 0.05 were considered statistically significant."

Furthermore, we removed the section about survival analysis from the results and revised Table 1 to be compatible with the analyses.

Reviewer #3, Comment 10:

Results

- Comorbidities: Why were the 6 patients without comorbidities in the hospital? What were the culture sites and why were these obtained?

Response to Reviewer #3, Comment 10: As you correctly mentioned, we had six patients categorized as without comorbidity, while they had comorbidities that were not mentioned in our list. Therefore, we added those to the comorbidities list and revised the results. Please see Response to Reviewer #3, Comment 6.

Reviewer #3, Comment 11:

- Microbiologic characteristics: were the two children included in the four patients who fulfilled criteria for persisting infection / colonization?

Response to Reviewer #3, Comment 11: Yes, we rephrased the sentence and provided more details about patients with persisting infections. Please see Response to Reviewer #3, Comment 4.

Reviewer #3, Comment 12:

- Microbiologic characteristics: did any patients have more than one positive culture?

Response to Reviewer #3, Comment 12: Yes, please see Response to Reviewer #3, Comment 11.

Reviewer #3, Comment 13:

- Microbiologic characteristics: please specify what antimicrobials were tested in what time period.

Response to Reviewer #3, Comment 13: Thank you for the suggestion. We agree that it is important to provide more details about antimicrobial susceptibility testing when all the isolates were not tested against the same antibiotics. Although, it is difficult to specify which antimicrobials were tested in what time period from 2002 to 2020. We tried to provide more information about antimicrobial susceptibility testing in Table 2, on page 18, lines 1-4. We also added on page 10, lines 4-6 which reads:

“More than 52, 51, 49, 49, and 47 the cultured *D. acidovorans* were susceptible to meropenem, ceftazidime, imipenem, ciprofloxacin, and piperacillin / tazobactam, respectively, while 37, 36, and 32 were resistant to tobramycin, gentamicin, and colistin (Table 2).”

Table 2: Antimicrobial susceptibility pattern of *Delftia acidovorans*. The term Intermediate may represent a non-susceptible phenotype. Not all isolates were tested against all antibiotics. Although, 36 (88%) out of 41 and 32 (62%) out of 52 isolates were resistant to gentamicin and colistin, respectively.

Susceptibility	Penicillin	Ampicillin	Mecillinam	AMC	TZP	Cefuroxime	Ceftriaxone	Cefpodoxime	Ceftazidime	Sulfamethizol	Trimethoprim	TMP-SMX	Gentamicin	Tobramycin	Tetracycline	Ciprofloxacin	Moxifloxacin	Nitrofurantoin	Imipenem	Meropenem	Chloramphenicol	Aztreonam	Colistin
Resistant	>44	50	18	<3	<3	21	8	3	<3	4	19	<3	36	37	<3	>5	<3	4	<3	3	<3	9	32
Intermediate	*	<3	*	<3	7	17	18	*	>3	*	16	*	4	14	<3	3	<3	3	3	*	*	29	13
Sensitive	*	<3	<3	8	47	14	18	8	51	42	13	32	<3	4	29	49	8	<3	49	>52	3	16	7
Not Tested	>13	7	>39	48	<5	7	15	>47	3	>12	11	>23	18	4	27	<3	48	>48	>4	>2	>49	5	>3

*Not reported

Abbreviations: **AMC**, Amoxicillin/ clavulanic acid; **TMP-SMX**, Trimethoprim-Sulfamethoxazole; **TZP**, Piperacillin / Tazobactam;

Reviewer #3, Comment 14:

Table 3 is not included in the paper for review.

Response to Reviewer #3, Comment 14: We are sorry for the typo error in Table 3; a subset of data about antimicrobial susceptibility testing was included in Figure 1. We removed Figure 1 and provided the complete data in Table 2. Please see the response to Response to Reviewer #3, Comment 13.

Reviewer #3, Comment 15:

- Outcomes: were any of the patients in the ICU at the time of culture? What was the timing of cultures relative to hospital admission?

Response to Reviewer #3, Comment 15: We mentioned that none of the patients were admitted to ICU within 30 days after infection. To clarify, we elaborated on page 10, lines 13-16, which read:

"None, five (8.5%), and eight (14%) of the 59 patients were admitted to ICU within 30, 180, and 365 days after the first positive culture, respectively. Four out of the 59 patients (7%) died within 30 days after the first positive culture, ten (17%) died within 180 days, and 15 (25%) patients died within the 365 days after the first positive culture. Only one patient died after having been admitted to ICU."

now reads:

"Thirty-one (53%) out of the 59 patients were admitted to hospital at the time of positive culture. Six (10%) were admitted from more than 30 days before the positive culture. Twelve (20%) patients were admitted within 1 to 30 days before, and 14 (24%) patients on the day of positive culture. None of the patients were admitted to ICU at the time of the positive culture."

We added on page 10, line 21-22, and page 11, lines 1-2:

"Among the four patients who died within 30 days after the first positive culture, the specimens were collected from airway secretions (n=1), tissue or wound (n=2), and medical equipment (n=1). Moreover, three out of the four had polymicrobial cultures."

Reviewer #3, Comment 16:

- Survival analysis: again, this should be removed. The study is not adequately designed or powered for any inferential statistics.

Response to Reviewer #3, Comment 16: We agree with your comment and removed the survival analyses.

Reviewer #3, Comment 17:

General

- The descriptive characteristics of the patients are incredibly limited and do not help with understanding the patient population included. Standard items, such as relevant medical

comorbidities (COPD, diabetes, etc.) in addition to more detailed information (solid organ transplantation, malignancy status, etc.) should be provided in Table 1. Additionally, microbiologic source and epidemiologic classification of the infection must be provided.

Response to Reviewer #3, Comment 17: We agree with your comment and provided more details as was suggested. In Supplementary Table 1, we reported comorbidities. Please see Response to Reviewer #3, Comment 6.

Reviewer #3, Comment 18:

- The description of susceptibility results is very much appreciated. However, Figure 2 should provide information on the number of isolates tested for each drug and include all drugs tested - this appears to be a subset.

Response to Reviewer #3, Comment 18: Thank you for raising this point. We provided more information in Table 2. Please see Response to Reviewer #3, Comment 13. The susceptibility results were discussed on page 11, lines 22-23 and page 12, lines 1-6.

Reviewer #3, Comment 19:

- No information at all is provided on how the patients were treated. The sample size is small, but as mentioned, this is the largest study to date. Some information on treatment should be included.

Response to Reviewer #3, Comment 19: Thank you for raising this point. We agree that the information about antibiotic therapy is important. The data was available for 35 out of the 59 patients. It was challenging to differentiate if the prescriptions were for *D. acidovorans* or other co-cultured bacteria. Therefore, in Supplementary Table 2, we provided information about antibiotics prescribed within 3 months before and within one week after the first positive *D. acidovorans*. Furthermore, we added on page 10, lines 4-10, which reads:

"Thirty-five (59%) out of the 59 patients had complete information about previous antibiotic therapy. Twenty-five (71%) of 35 received at least one antibiotic within three months before the first positive D. acidovorans. Penicillins and fluoroquinolones were patients' most common antibiotics before a positive D. acidovorans.

After a positive D. acidovorans, 23 (66%) of 35 patients received at least one new antibiotic. Fluoroquinolones and meropenem were the antibiotics most prescribed after a positive D. acidovorans (Supplementary Table 2)."

Supplementary Table 2: Number of patients who received antibiotic within before or after the first positive *D. acidovorans*.

Antibiotic	Number of patients who received antibiotic within 3 months before the first positive D. acidovorans	Number of patients who received a new antibiotic after the first positive D. acidovorans *
Penicillin or Ampicillin	3	<3
Amoxicillin	6	3
Amoxicillin / Clavulanic acid	<3	-
Dicloxacillin	3	-
Pivmecillinam	<3	-
Piperacillin/Tazobactam	3	3
Cefuroxime	<3	<3
Doxycycline	<3	-
Linezolid	<3	-
Vancomycin	3	<3
Clarithromycin or Azithromycin	6	-
Trimethoprim / Sulfamethoxazole	3	<3
Moxifloxacin or Ciprofloxacin	10	9
Nitrofurantoin	<3	-
Meropenem	4	8
Gentamicin or Tobramycin	4	-
Colistin	4	<3
Rifampicin	<3	-
Metronidazol	<3	-
*If the antibiotic was started before the positive D. acidovorans was not counted here. - No information was available.		

Reviewer #4, Comment 1:

The manuscript by Hojgaard et al provides an interesting look at the prevalence and clinical outcomes associated with *Delftia acidovorans* infections during a specific timeframe in a high resource setting. Although there are clear limitations to the retrospective study, the authors clearly outline these in their discussion. Some specific comments:

Lines 161-166: The manuscript comments on using both CLSI and EUCAST breakpoints for the organism; it is unclear as to why this was the case.

Response to Reviewer #4, Comment 1: Thank you for this question, we understand why this may be difficult to understand. Use of both CLSI and EUCAST breakpoints is because the standard method in our center changed in 2016. We revised this. Please see Response to Reviewer #3, Comment 7.

Reviewer #4, Comment 2:

Line 196: Were all cases felt to represent infection, especially cultures isolated from superficial wounds? This reviewer has seen this organism in polymicrobial wounds and

sputum cultures where it likely has represented a colonizer and not necessarily clinically significant.

Response to Reviewer #4, Comment 2: We agree that some of the positive cultures could be colonizations without clinical significance. Unfortunately, it was not possible to differentiate between colonization and infection in this retrospective study. We defined infection on page 6, line 17-18, which reads:

"D. acidovorans infection was defined as the growth of D. acidovorans in clinical samples obtained from any site in a patient either admitted to the hospital or attending an outpatient consultation."

We also added in the limitations, on page 12, lines 20-22 which read:

"Moreover, we had complete information on death and laboratory information on antimicrobial susceptibility. We obtained permission to review clinical charts from deceased patients, but alive patients should provide written consent, and not all of them accepted to participate. Therefore, we had some degree of bias in selecting cases, with a potential to overestimate the contribution of comorbidities in the population described. We also had to address missing data, as not all historic patient's charts and deceased patient's charts were available to access digitally via electronic records."

now reads

"Moreover, we had complete information on death and laboratory information on antimicrobial susceptibility. We obtained permission to review clinical charts from deceased patients, but alive patients should provide written consent, and not all of them accepted to participate. Therefore, we had some degree of bias in selecting cases, potentially overestimating the contribution of comorbidities in the population described. We also had to address missing data, as not all historic patient's charts and deceased patient's charts were available to access digitally via electronic records. Furthermore, using current data, it was not possible to differentiate between colonization and infection. Therefore, it is possible that some patients had colonizations without clinical importance."

Reviewer #4, Comment 3:

With the small numbers included in the study, it would have been possible to perform a chart review to assess the cause of mortality in the cases that were deceased. How many cases can be reasonably attributed to Delftia infection?

Response to Reviewer #4, Comment 3: We agree that this is an important point. Although, due to the simultaneous comorbidities and co-cultured bacteria, it is challenging to attribute death to Delftia infection. Unfortunately, the cause of death is registered in the Danish Register of Causes of Death, and we did not have permission to access this registry.

Reviewer #5 (Comments for the Author):

Reviewer #5, Comment 1:

General Comments

1. Evaluating the number of comorbidities is certainly of interest, however, "equal footing" for each of the comorbidities evaluated may not be optimal. Consideration to utilization of a more comprehensive score such as the Charlson or Elixhauser Comorbidity Indexes should be considered.

Response to Reviewer #5, Comment 1: Thank you for your suggestion. We decided to use Charlson Comorbidity Index when designing the study. However, it was impossible to calculate a valid score due to the missing data. Therefore we reported the number of comorbidities instead.

Reviewer #5, Comment 2:

2. Due to the high number of polymicrobial infections (70%) differentiating the role/impact of *D. acidovorans* is extremely challenging. This will be a major limitation regardless of how the data is analyzed and presented. However, it would be helpful to the readership to recognize this as a limitation and attempt to be as detailed as possible when depicting the types of infections that were polymicrobial. For instance, more detailed summary of suspected source of polymicrobial bloodstream infections may assist in identifying opportunities for recognition and optimal treatment in similar patients. Moreover, the specific organisms identified concomitantly suggest a patient population that has likely had significant healthcare and possibly antibiotic exposure. Information on the hospitalization (previous or time from admission to culture collection) and any antibiotic exposure would add substantial value to understanding patients at risk for infections with *D. acidovorans*, please consider including if possible.

Response to Reviewer #5, Comment 2: Thank you for the comment. We provided more information about previous and current antibiotic exposure, antibiotic susceptibility, the timing of hospital admission, and the outcomes. Please see Response to Reviewer #3, Comments 4, 13, 15, and 19.

Reviewer #5, Comment 3:

3. The finding of lower mortality among female patients may be of interest, however, context is needed. Is there biologic plausibility for such a finding? Related to infection type differences between females/males? Or is this simply a chance finding. Context in the discussion is needed if this is to remain an emphasized result.

Response to Reviewer #5, Comment 3: Thank you for the suggestion. The number of patients was low. Therefore, we were not able to investigate the reasons for lower one-year survival in female patients. Other respected reviewers emphasized against using survival analyses due to the small sample size and missing data; therefore, we chose to remove both Cox regression and survival probabilities and reported the proportions instead.

Reviewer #5, Comment 4:

4. Because mortality at 365 days could be attributable to numerous confounders, consideration of evaluating other clinical outcomes may be warranted. Evaluation of "clinical failure" or "infection-related mortality" may be of interest in more specifically evaluating the impact of the D. acidovorans infection on the mortality.

Response to Reviewer #5, Comment 4: As you correctly pointed out, one-year mortality could be attributable to numerous confounders. Our patients had comorbidities, and the cultures were polymicrobial. Therefore, we chose to remove survival probabilities. Please see Response to Reviewer #4, Comment 3, and Response to Reviewer #5, Comment 3.

Reviewer #5, Specific Comment 1:

Specific Comments

1. P6 L125: Please consider specifying here whether after second contact non-responders were excluded (as it seems they were from results later in the manuscript).

Response to Reviewer #5, Specific Comment 1: Thank you for pointing this out. Yes, we exclude non-responders. Please see Response to Reviewer #3, Comment 5.

Reviewer #5, Specific Comment 2:

2. P7 L150-154: Were transplant (solid organ) or other immunocompromising comorbidities or therapies considered?

Response to Reviewer #5, Specific Comment 2: Yes, please see Response to Reviewer #3, Comment 6.

Reviewer #5, Specific Comment 3:

3. P8 L161-166 and 19 Figure 2: Changes in testing methodology and possibly breakpoints utilized over time could influence the susceptibility results depicted. Please consider expressly identify the frequency in which testing modalities were utilized. Additionally, consideration for more directly sharing MIC results or what specific breakpoint was utilized for interpretation is needed to allow the readership to interpret such results in a clinical context.

Response to Reviewer #5, Specific Comment 3: Thanks for the suggestion. We agree that changes in susceptibility testing methodology and possibly breakpoints could influence the susceptibility results. Unfortunately, we did not have the MIC values to report in this manuscript. We added in methods that "All susceptibility testing, identification, and other testing were done in accordance with routine laboratory practices." Please see Response to Reviewer #3, Comments 7 and 13.

Moreover, we added in the limitations on page 12, lines 22-23, and page 13, lines 1-2 which reads:

"Changes in susceptibility testing methods and possibly breakpoints could influence the susceptibility results. Although everything was done according to the standard of care and methods were calibrated if necessary."

Reviewer #5, Specific Comment 4:

4. P16-17 Tables 1 and 2 (and results section): Infection type (or specimen source) findings are interesting with lower mortality rate in bloodstream vs urinary source infections. Perhaps this relates to general comment #4 above (infection related mortality or due to comorbidity/confounder?). More detail surrounding the infection types and how they interact with the outcome (mortality) would seemingly be of utility to the readership.

Response to Reviewer #5, Specific Comment 4: We agree and added more details. Please see Response to Reviewer #3, Comment15.

Reviewer #5, Specific Comment 5:

5. P11 L237-238: This statement seems to lack context: is it compared to no infection or other pathogen, please consider providing more context.

Response to Reviewer #5, Specific Comment 5: We agree that the statement needs more context. Due to the change in analyses, we removed this sentence.

Reviewer #5, Specific Comment 6:

6. P16 Table 1: Specifics on the individual comorbidities (e.g., COPD, CF) and outcomes would be of interest to readership, please consider including in the table.

Response to Reviewer #5, Specific Comment 6: Thank you for the suggestion. Please see Response to Reviewer #3, Comment 6.

June 27, 2022

Dr. Zitta Barrella Harboe
Copenhagen University Hospital, North Zealand
Copenhagen
Denmark

Re: Spectrum00326-22R1 (Characteristics and Outcomes of Patients with Delftia acidovorans Infections: A Retrospective Cohort Study)

Dear Dr. Zitta Barrella Harboe:

Your manuscript has been accepted, and I am forwarding it to the ASM Journals Department for publication. You will be notified when your proofs are ready to be viewed.

Sincerely,

Krisztina Papp-Wallace
Editor, Microbiology Spectrum

Journals Department
Supplementary Tables: Accept